# PEDF-Mediated Mitophagy Triggers the Visual Cycle by Enhancing Mitochondrial Functions in a H_2_O_2_-Injured Rat Model

**DOI:** 10.3390/cells10051117

**Published:** 2021-05-06

**Authors:** Jae Yeon Kim, Sohae Park, Hee Jung Park, Se Ho Kim, Helen Lew, Gi Jin Kim

**Affiliations:** 1Department of Biomedical Science, CHA University, Seongnam 13488, Korea; janejykim92@gmail.com (J.Y.K.); sohae11@snu.ac.kr (S.P.); heejung970328@gmail.com (H.J.P.); sehokim93@gmail.com (S.H.K.); 2Research Institute of Placental Science, CHA University, Seongnam 13488, Korea; 3CHA Bundang Medical Center, Department of Ophthalmology, CHA University, Seongnam 13496, Korea; eye@cha.ac.kr

**Keywords:** mitophagy, pigment epithelium-derived factor, placenta-derived mesenchymal stem cells, visual cycles, retinal degenerative diseases

## Abstract

Retinal degenerative diseases result from oxidative stress and mitochondrial dysfunction, leading to the loss of visual acuity. Damaged retinal pigment epithelial (RPE) and photoreceptor cells undergo mitophagy. Pigment epithelium-derived factor (PEDF) protects from oxidative stress in RPE and improves mitochondrial functions. Overexpression of PEDF in placenta-derived mesenchymal stem cells (PD-MSCs; PD-MSCs^PEDF^) provides therapeutic effects in retinal degenerative diseases. Here, we investigated whether PD-MSCs^PEDF^ restored the visual cycle through a mitophagic mechanism in RPE cells in hydrogen peroxide (H_2_O_2_)-injured rat retinas. Compared with naïve PD-MSCs, PD-MSCs^PEDF^ augmented mitochondrial biogenesis and translation markers as well as mitochondrial respiratory states. In the H_2_O_2_-injured rat model, intravitreal administration of PD-MSCs^PEDF^ restored total retinal layer thickness compared to that of naïve PD-MSCs. In particular, PTEN-induced kinase 1 (PINK1), which is the major mitophagy marker, exhibited increased expression in retinal layers and RPE cells after PD-MSC^PEDF^ transplantation. Similarly, expression of the visual cycle enzyme retinol dehydrogenase 11 (RDH11) showed the same patterns as PINK1 levels, resulting in improved visual activity. Taken together, these findings suggest that PD-MSCs^PEDF^ facilitate mitophagy and restore the loss of visual cycles in H_2_O_2_-injured rat retinas and RPE cells. These data indicate a new strategy for next-generation MSC-based treatment of retinal degenerative diseases.

## 1. Introduction

Retinal degenerative diseases are heterogeneous conditions and include diabetic retinopathy and age-related macular degeneration (AMD) [1]. Loss of the visual cycle and photoreceptor damage in AMD are the part of causes of blindness due to mitochondrial dysfunction [2,3]. The stages of AMD include hard drusen, which is associated with localized malfunctions of the retinal pigment epithelium, soft drusen associated with diffuse dysfunction of the retinal pigment epithelium, and damage to photoreceptors [4]. In retinal degeneration, pigment epithelium-derived factor (PEDF), which is produced in retinal pigment epithelial (RPE) cells and maintains retinal homeostasis, is decreased, while the expression of vascular endothelial growth factor (VEGF) is increased [5,6].

PEDF is a 50-kDa secreted glycoprotein known as serine proteinase inhibitor (serpin) F1 and is synthesized by cultured RPE cells [7]. In the human retina, PEDF localizes to photoreceptor cells, the inner nuclear layer (INL), the ganglion cell (GC) layer, the inner plexiform layer (IPL), and pigment epithelial cells (PECs). PEDF plays important roles in protecting retinal functions [8]. Deficiencies or defects in PEDF are intimately associated with the progression of angiogenic diseases such as diabetic retinopathy [9]. In a diabetic mouse model, PEDF has antioxidant effects that protect against oxidative stress by promoting cell survival and reducing pathological angiogenesis [10]. Moreover, PEDF treatment prevents oxidative stress by improving mitochondrial structure and function and activating the PI3K/AKT and MAPK pathways in H_2_O_2_-injured RPE cells [11].

The classical cycle is involved in the cycling of retinoids from the rod outer segment and the retinal pigment epithelium [12,13,14]. All-trans retinal is released from opsin and decreased by all-trans retinol dehydrogenase (at-RDH). In retinal degeneration, a lack of the PEDF receptor induces the regressive survival of photoreceptor cells [15]. In cone photoreceptor cell damage, PEDF signaling through the PEDF receptor induces marked increases in S cone numbers and survival [16]. Downregulated PEDF levels in RPE cells decrease the expression of LRAT and RDH11. However, the effect of PEDF on the visual cycle in H_2_O_2_-damaged RPE cells is still unknown.

Autophagy is a programmed and evolutionarily conserved catabolic pathway that degrades excessive or damaged organelles and cellular proteins through the formation of autophagosomes [17]. Additionally, specific autophagy processes have been reported, including mitophagy, that are mediated by PINK1/PARKIN signaling and selectively remove damaged or excessive mitochondria [18]. In AMD-like pathology, oxidative stress-induced RPE cells activate mitochondrial clearance by a mitophagic mechanism [19]. Knockdown of PINK1 expression downregulated phosphorylated PARKIN levels in RPE cells treated with H_2_O_2_ [20]. In addition, the effect of PEDF expression on mitophagy in RPE cells is still unclear.

Previously, a mitochondrial oxidative stress model was established using H_2_O_2_ in rats based on this study [21]. We found that overexpressed PEDF in PD-MSCs (PD-MSCs^PEDF^) improved antioxidant effects and mitochondrial biogenesis associated with retinal regeneration in a H_2_O_2_-injured rat model and RPE cells [22]. In this study, we focused on recovery of the visual cycle through mitophagy and mitochondrial functions by PD-MSC^PEDF^ in a H_2_O_2_-injured rat model and RPE cells.

## 2. Materials and Methods

### 2.1. Cell Culture and Gene Transfection

Placentas were collected from healthy women (37 gestational weeks) by the Institutional Review Board of CHA Gangnam Medical Center, Seoul, Korea (IRB-07-18). PD-MSCs were isolated and incubated as previously described in α-modified minimal essential medium (α-MEM; HyClone, Logan, UT, USA) containing 10% fetal bovine serum (FBS; Gibco, Carlsbad, CA, USA), 1% penicillin/streptomycin (P/S; Invitrogen, Carlsbad, CA, USA), 25 ng/mL human fibroblast growth factor-4 (FGF-4; PeproTech, Rocky Hill, NJ, USA), and 1 μg/mL heparin (Sigma-Aldrich, St. Louis, MO, USA). The PEDF plasmid was constructed as previously described. To induce overexpression of the PEDF gene, naïve PD-MSCs (passage = 7) were transfected with the PEDF plasmid based on nonviral 4D nucleofection (Lonza, Basel, Switzerland) according to our previous reports [23]. After 24 h of transfection, the cells were selected using 200 μg/mL hygromycin. Human retinal pigment epithelium-derived ARPE-19 (ATCC, Manassas, VA, USA) was maintained in Dulbecco’s modified Eagle medium (DMEM; Gibco) containing 10% FBS (Gibco) and 1% P/S (Invitrogen). Cells were cultured at 5% CO_2_ and 37 °C.

### 2.2. In Vitro Coculture System

To analyze the effects of naïve PD-MSCs or PD-MSCs^PEDF^, ARPE-19 cells were treated with hydrogen peroxide (H_2_O_2_; 200 μM; Sigma-Aldrich) for 2 h and cocultured with naïve PD-MSCs or PD-MSCs^PEDF^ (5 × 10^3^/cm^2^) in 8-μm pore Transwell inserts (Corning, NY, USA) in α-MEM (HyClone) containing 1% P/S (Invitrogen) for 24 h at 5% CO_2_ and 37 °C.

### 2.3. Animals and MSC Transplantation

Seven-week-old male Sprague-Dawley rats (Orient Bio Inc., Seongnam, Korea) were maintained in an air-conditioned facility. All rats were anesthetized by 75 mg/mL of tribromoethanol (avertin; Sigma-Aldrich) by intraperitoneal injection. Acute eye disease was induced by a intravitreally single injection of H_2_O_2_ (10 μg/μL; Sigma-Aldrich) for 2 weeks, and rats in the sham group were injected with balanced salt solution (BSS). The rats were divided into the following groups: the sham (sham; *n* = 6), which received H_2_O_2_ injection for 2 weeks (H_2_O_2_; *n* = 6); naïve PD-MSCs (PD-MSCs; 2 × 10^5^; *n* = 6); and PD-MSCs^PEDF^ (PD-MSCs^PEDF^; 2 × 10^5^; *n* = 6). A week after MSC transplantation, the rats were euthanized using a CO_2_ chamber, body and tissue weights were measured, and eyeballs and serum were collected. The experimental protocol was approved by the Institutional Animal Care and Use Committee of CHA University, Seongnam, Korea (IACUC-200033).

### 2.4. Histopathological and Immunofluorescence Analysis

Enucleated eyeballs were examined and fixed after immersion with neutral buffered formalin (NBF) from each group (*n* = 6). Each sample was embedded in paraffin and cut into 5-μm-thick sections in the central region on the optic disc of the eyeball for hematoxylin and eosin (H&E) staining and immunohistochemistry (IHC). To analyze the genes in retinal layers of the eyeball, each eyeball section was stained with anti-VEGF (1:200; Novus Biologicals, CO, USA), anti-PEDF (1:200; LSbio, Washington, DC, USA), anti-LC3 (1:200; Cell Signaling Technology, Danvers, MA, USA), anti-PINK1 (1:100; Abcam, Cambridge, UK), and anti-RDH11 (1:200; Bioss, Woburn, MA, USA) antibodies. Horseradish peroxidase-conjugated streptavidin–biotin complex (DAKO, Santa Clara, CA, USA) and 3,3-diaminobenzidine (DAKO) were used to generate chromatic signals. Morphometric images of 4–5 whole sections of eyeballs from each group were used to quantify positive IHC signals and determine the lengths of retinal layers using the HistoQuant program of the slide scanner (Pannoramic P250, 3DHISTECH, Ltd., Budapest, Hungary). To confirm the expression of mitophagy genes in ARPE-19, the cells were fixed with 4% paraformaldehyde (PFA) in phosphate-buffered saline (PBS) and blocked in blocking solution (DAKO) in the dark. Anti-PINK1 (1:100; Abcam), anti-RDH11 (1:200; Bioss), and mtTracker (50 nM; Invitrogen) were added and incubated at 4 °C overnight. Alexa Fluor™ 594 goat anti-rabbit IgG (1:250; Invitrogen) secondary antibodies were used, (Appendix A) and nuclei was stained with 4′6-diamidino-2-phenylindole (DAPI; Invitrogen). The images were observed by confocal microscopy (LSM880) and analyzed using ZEN blue software (ZEISS, Oberkochen, Germany).

### 2.5. Quantitative Real-Time Polymerase Chain Reaction (qRT-PCR)

Total RNA was extracted from rat retinas or cells using TRIzol LS (Invitrogen) based on the manufacturer’s method. cDNA was synthesized using the SuperScript III reverse transcriptase (Invitrogen). qRT-PCR was assessed on a CFX Connect™ Real-Time System (Bio-Rad, Hercules, CA, USA), with primers (Appendix A) and SYBR Green PCR master mix (Roche, Basel, Switzerland). Gene expression was quantified by the 2-ΔΔCT method, and all data were analyzed in triplicate.

### 2.6. Western Blotting

To assess the expression of autophagy and mitophagy proteins in ARPE-19 cells, the samples were lysed in lysis buffer (Sigma-Aldrich) with a phosphatase inhibitor (AG Scientific Inc., San Diego, CA, USA) and a protease inhibitor cocktail (Roche). Protein lysates were separated by sodium dodecyl sulfate-polyacrylamide gel electrophoresis (SDS-PAGE) and transferred to PVDF membranes. Primary antibodies were used as follows: anti-PINK1 (1:500; Abcam), anti-ATG7 (1:500; Santa Cruz Biotechnology, Dallas, TX, USA), anti-Beclin1 (1:1000; Santa Cruz Biotechnology), anti-LC3 (Cell Signaling Technology), and anti-GAPDH (1:3000; AbFrontier, Seoul, Korea). The following secondary antibodies were used: anti-HRP-conjugated mouse IgG (1:5000; Bio-Rad) and anti-HRP-conjugated rabbit IgG (1:5000; Bio-Rad) (Appendix A). Each band was subject to chemiluminescent detection using ECL reagent (Bio-Rad) and quantified using Image J software (NIH, Bethesda, MD, USA).

### 2.7. XF Mito Stress Assay

To analyze mitochondrial metabolic conditions in live naïve PD-MSCs and PD-MSCs^PEDF^, the oxygen consumption rate (OCR) and extracellular acidification rate (ECAR) were assessed using an XF24 Extracellular Flux Analyzer (Seahorse Bioscience, North Billerica, MA, USA). Naïve PD-MSCs and PD-MSCs^PEDF^ (7 × 10^3^ cells/well) were cultured in XF 24-well microplates and equilibrated in XF buffer for 60 min by repeated cycles of mixing for 3 min, incubation for 2 min, and measurement for 30 min in non-CO_2_ conditions. Each cell type was sequentially treated with 0.5 μM oligomycin, 0.5 μM carbonyl cyanide-p-trifluoromethoxyphenylhydrazone (FCCP), and a 1 μM rotenone/antimycin A (AA) mixture according to the manufacturer’s recommendation by the Seahorse XF24 Analyzer. The OCR/ECAR was normalized to the total cell number.

### 2.8. Enzyme-Linked Immunosorbent Assay (ELISA)

To confirm PEDF overexpression in PD-MSCs, human PEDF in cell culture supernatant was analyzed according to the instructions using the human PEDF ELISA kit (Abcam). IL-6 and IL-10 levels in rat serum (*n* = 6/group) were analyzed by rat ELISA kit (Abcam) to determine proinflammatory and anti-inflammatory factor levels. Samples and standards were added to appropriate wells. Then, each antibody was incubated at room temperature. These were aspirated and each well was washed four times. TMB solution was incubated and stop solution was added. The OD at 450 nm was measured by an Epoch microplate reader (BioTek, Winooski, VT, USA). Each concentration was assessed by the trend line equation. All reactions were performed in triplicate.

### 2.9. Statistical Analysis

All data were analyzed using GraphPad Prism version 9.0 (GraphPad Software, San Diego, CA, USA), and statistically significant differences were assessed using two-tailed unpaired Student’s t-tests or one-way analysis of variance (ANOVA), followed by an appropriate post hoc test. P values less than 0.05 were considered statistically significant.

## 3. Results

### 3.1. PD-MSCs^PEDF^ Enhance Mitochondrial Activity

We confirmed endogenous PEDF levels in the culture supernatants of PD-MSCs^PEDF^. Compared to naïve PD-MSCs, PD-MSCs^PEDF^ exhibited significantly increased secretion (25-fold) (Figure 1A; *p* < 0.05). The mRNA expression of PEDF was markedly increased (Figure 1B; *p* < 0.05). To verify mitochondrial respiration in PD-MSCs^PEDF^, we analyzed the OCR and ECAR of live cells by an XF analyzer. PD-MSCs^PEDF^ showed more energetic conditions than naïve PD-MSCs (Figure 1C). In the presence of oligomycin, mitochondrial ATP production in PD-MSCs^PEDF^ was superior to that in naïve PD-MSCs. Maximal respiration levels in PD-MSCs^PEDF^ were increased by treatment with the mitochondrial oxidative phosphorylation uncoupler FCCP. Rotenone/antimycin A treatment enhanced spare cellular capacity (Figure 1D). Moreover, the mRNA expression of mitochondrial biogenesis markers (e.g., sirtuin-1 (SIRT-1), peroxisome proliferator-activated receptor gamma coactivator 1 alpha (PGC1α), nuclear respiratory factor 1 (NRF1), estrogen-related receptor alpha (ERRα), and mitochondrial transcription factor A (TFAM)) was markedly increased in PD-MSCs^PEDF^ (Figure 1E). In addition, uncoupling proteins 2 and 3 (UCP2 and 3, respectively), which attenuate mitochondrial ROS [24], showed increased gene expression in PD-MSCs^PEDF^ compared to naïve PD-MSCs (Appendix A). Additionally, the process of mitochondrial translation is critical for mitochondrial biogenesis [25]. We confirmed the mitochondrial translational phase by qRT-PCR analysis. Compared to naïve PD-MSCs, PD-MSCs^PEDF^ exhibited significantly increased expression of mitochondrial initiation factors 2 and 3 (MTIF2 and MTIF3, respectively), mitochondrial translation release factor 1 (MTRF1), mitochondrial ribosome recycling factor (MRRF), and mitochondrial elongation factor 1 (MIEF1) (Figure 1F; *p* < 0.05). These results suggest that PD-MSCs^PEDF^ improve mitochondrial respiratory conditions by enhancing mitochondrial biogenesis and translation compared with those of naïve PD-MSCs.

### 3.2. PD-MSC^PEDF^ Transplantation Restores Retinal Function in a H_2_O_2_-Injured Rat Model

To induce an acute retinal degeneration model, H_2_O_2_ (10 μg/μL) injection was conducted for 1 and 2 weeks, while the sham group was injected with the same volume of BSS. Naïve PD-MSCs (H_2_O_2_ + Naïve) and PD-MSCs^PEDF^ (H_2_O_2_ + PEDF+) were intravitreally injected into rats. After 1 and 2 weeks, eyeball weight was significantly decreased in the H_2_O_2_-injured groups (Appendix A). However, naïve PD-MSC transplantation induced a slight increase. Interestingly, the PD-MSCs^PEDF^ group showed a significant difference (Figure 2A; *p* < 0.05). We analyzed inflammatory cytokines in rat serum and confirmed proinflammatory IL-6 and anti-inflammatory cytokine IL-10 levels. After H_2_O_2_ treatment, increased IL-6 and decreased IL-10 were observed compared with those in the sham group. After transplantation of naïve PD-MSCs or PD-MSCs^PEDF^, decreases in IL-6 and increases in IL-10 levels were assessed. In particular, IL-10 in the PD-MSCs^PEDF^ group was markedly greater than that in the naïve PD-MSCs group (Figure 2B,C; *p* < 0.05). We further evaluated histological structural changes in retinal layers in the eyeball using H&E staining. Following H_2_O_2_ injection, total retinal layers showed abnormal conditions and neutrophil infiltration.

Interestingly, the administration of PD-MSCs^PEDF^ preserved the integrity of the retinal layers (Figure 2D). Additionally, the thickness of the total retina was reduced in the H_2_O_2_ group compared to the sham group. Other layers, including the ganglion cell layer (GCL) and inner plexiform layer (IPL), were damaged. The effects in the naïve PD-MSCs and PD-MSCs^PEDF^ groups were restored. In particular, PD-MSC^PEDF^ transplantation increased the length of the total retina layer (Figure 2E). In detail, the thicknesses of the outer nuclear layer (ONL) and inner nuclear layer (INL) were assessed. The ONL, which contains the nuclei of the cone and rod photoreceptor, and the INL, which modulates synaptic connections, are important for visual cycles. The thicknesses of the ONL and INL were increased in the PD-MSC^PEDF^ group compared to the naïve PD-MSC group (Figure 2F,G, *p* < 0.05). These data suggest that PD-MSC transplantation induced an anti-inflammatory effect on a H_2_O_2_-injured retinal degeneration in the rat model and that the administration of PD-MSCs^PEDF^ structurally enhanced retinal restoration in a H_2_O_2_-induced retinal degeneration rat model.

### 3.3. PD-MSCs^PEDF^ Balanced VEGF and PEDF Levels in a H_2_O_2_-Induced Rat Model

Retinal degenerative diseases, including AMD, are characterized by the growth of abnormal blood vessels, resulting in increased levels of the angiogenic factor VEGF and decreased levels of the antiangiogenic factor PEDF [26]. We confirmed the mRNA expression of VEGF and PEDF in the rat retina. After H_2_O_2_ treatment, the expression of VEGF was highly increased, while the expression of PEDF was significantly decreased. Compared to naïve PD-MSC transplantation, PD-MSCs^PEDF^ reduced VEGF levels and enhanced PEDF levels (Figure 3A; *p* < 0.05). The mRNA expression of these two genes showed a negative correlation value of R^2^ = −0.7575 (Figure 3B). Moreover, endogenous VEGF and PEDF expression was analyzed by immunohistochemistry. Representative images showed the same trend as that of the mRNA expression (Figure 3C). The VEGF- and PEDF-positive areas were measured in each group. H_2_O_2_ treatment significantly increased the intensity of VEGF, whereas it decreased that of PEDF. After naïve PD-MSC and PD-MSC^PEDF^ transplantation, decreased VEGF-positive areas and increased PEDF-positive areas were observed. In particular, PD-MSCs^PEDF^ significantly improved PEDF expression in retinal layers (Figure 3D,E). Endogenous gene expression levels of VEGF and PEDF in the retina exhibited a negative correlation value of R^2^ = −0.8349 (Figure 3F). These results indicated that VEGF and PEDF expression were modulated by PD-MSC^PEDF^ transplantation in H_2_O_2_-injured rat retinas.

### 3.4. PD-MSCs^PEDF^ Improved Visual Cycles in Rat Retinal Layers and RPE Cells

The structural relationship between the retinal pigment epithelium and photoreceptors is critical for the visual cycle [12]. In RPE cells, all-trans retinol is converted into 11-cis retinal and is transferred to cellular retinoid-binding protein (RBP). All-trans-retinol is delivered by lecithin retinol acyl transferase (LRAT). Then, retinol dehydrogenase (RDH) 5 and 11 are involved in the oxidation of 11-cis retinol.

The generated 11-cis retinal crosses the interphotoreceptor matrix and enters photoreceptor cells. All-trans retinal is converted into all-trans-retinol by RDH12, 13, and 14 (Figure 4A). We analyzed the expression of RDH11, which is a key enzyme in the visual cycle, in the rat retina by IHC. Interestingly, the H_2_O_2_ injection group exhibited decreased gene expression. On the other hand, RDH11 levels in the naïve PD-MSC and PD-MSC^PEDF^ groups were significantly restored. In particular, the administration of PD-MSCs^PEDF^ significantly increased the expression levels of visual cycle enzymes (e.g., RDH11, 12, 13, and 14) compared to that of naïve PD-MSCs (Figure 4C; *p* < 0.05).

To further analyze visual enzymes in RPE cells, H_2_O_2_ stimulation was conducted for 2 h to induce oxidative stress, and naïve PD-MSCs or PD-MSCs^PEDF^ were indirectly cocultured with ARPE-19 cells for 24 h. Previously, we confirmed the gene expression of retinal function markers in ARPE-19 cells in response to PD-MSCs^PEDF^ [22]. We further analyzed the major enzymes involved in the visual cycle in RPE cells. The mRNA expression of RDH5 and RDH11 in H_2_O_2_-treated cells was reduced, as shown by qRT-PCR and immunofluorescence analysis (Figure 4D,E). Compared to coculture with naïve PD-MSCs, coculture with PD-MSCs^PEDF^ induced significant levels of RDH5 and RDH11. In addition, LRAT and RBP1 expression were also restored after coculture with PD-MSCs^PEDF^ compared to naïve PD-MSCs (Figure 4F,G; *p* < 0.05). These results indicate that the administration of PD-MSCs^PEDF^ improves visual cycles in H_2_O_2_-injured rat retinas and RPE cells.

### 3.5. The Administration of PD-MSCs^PEDF^ Induces Mitophagy in H_2_O_2_-Injured Rat Retinas

In retinal degenerative diseases, PEDF results in elevated autophagy in RPE cells [6]. Previously, our reports showed that PD-MSCs^PEDF^ enhanced antioxidant effects and mitochondrial functions in RPE cells [22]. We further analyzed mitochondrial autophagy, which is known as mitophagy, in H_2_O_2_-injured rat retinas. The mRNA expression of autophagy markers (e.g., microtubule-associated proteins 1A/1B light chain 3B (LC3), autophagy-related 7 (ATG7), and Beclin1 (BECN1)) was increased in the H_2_O_2_ group compared with the sham group. Additionally, the expression of autophagy-related markers after PD-MSC^PEDF^ transplantation was significantly different from that after naïve PD-MSC transplantation (Figure 5A; *p* < 0.05). In the rat retina, PD-MSCs^PEDF^ induced predominant LC3 expression, as shown by IHC (Figure 5B). Furthermore, mitophagy markers (e.g., mitofusin 2 (MFN2), PTEN-induced kinase 1 (PINK1), and Parkin RBR E3 ubiquitin protein kinase (PARKIN)) were analyzed by qRT-PCR. The expression of these factors was also increased in the H_2_O_2_ group. Compared to the naïve PD-MSCs group, the PD-MSCs^PEDF^ group showed significant enhancement (Figure 5C; *p* < 0.05). Endogenous PINK1 expression in rat retinal sections was confirmed by IHC. Compared to naïve PD-MSC transplantation, PD-MSC^PEDF^ transplantation also increased expression patterns (Figure 5D). These findings indicate that PD-MSCs^PEDF^ promote mitophagy in a PINK1-dependent manner in the rat retina.

### 3.6. Coculturing RPE Cells with PD-MSCs^PEDF^ Enhances Mitophagy

In H_2_O_2_-injured rat retinas, improvements in PINK1-dependent mitophagy occurred. Additionally, autophagy and mitophagy in H_2_O_2_-damaged RPE cells were analyzed after the cells were cocultured with naïve PD-MSCs or PD-MSCs^PEDF^. The mRNA expression of LC3 and ATG7 was decreased in H_2_O_2_-induced ARPE-19 cells. After coculture with naïve PD-MSCs or PD-MSCs^PEDF^, the expression of these genes was elevated. In particular, PD-MSCs^PEDF^ induced higher levels than naïve cells (Figure 6A; *p* < 0.05). In addition, the mitophagy-related genes PINK1 and PARKIN tended to have the same expression patterns (Figure 6B). To further analyze PINK1 levels in the mitochondria of RPE cells, immunostaining with PINK1 and mtTracker was performed in ARPE-19 cells.

H_2_O_2_ treatment induced PINK1 expression, whereas mitochondrial signals were decreased. However, coculture with naïve PD-MSCs slightly decreased PINK1 expression and increased mtTracker signals. In particular, PD-MSCs^PEDF^ significantly improved both signals by increasing PINK1 expression (Figure 6C). In addition, the protein expression of LC3 type 2 and Beclin1 tended to be increased in PD-MSCs^PEDF^. After H_2_O_2_ treatment, PINK1 and ATG7 expression was induced in RPE cells compared with normal control cells; however, naïve PD-MSCs did not mediate this expression. In particular, coculture with PD-MSCs^PEDF^ significantly improved PINK1 and ATG7 levels (Figure 6D,E, Appendix A). These data suggest that PD-MSC^PEDF^ coculture enhances mitophagy signaling in H_2_O_2_-damaged RPE cells.

## 4. Discussion

The treatment of retinal degenerative diseases is ongoing in the developed world. Approximately 42 phase 1–3 clinical trials for retinal degenerative disease treatment have been conducted (www.clinicaltrials.gov (accessed on 22 March 2021)). As standard treatments, ranibizumab and bevacizumab, which are monoclonal antibodies that inhibit angiogenesis by targeting VEGF, are usually used for drug therapy. The outcomes indicated changes in the visual acuity scores of patients with AMD or diabetes. However, these therapies have several side effects, including conjunctival hemorrhage, intraocular inflammation, and increased intraocular pressure [27]. Additionally, several trials are necessary to assess long-term observations in an ongoing follow-up trial phase and continuous improvements in the relief of symptoms.

Currently, stem cell therapy has been applied for acute and chronic retinal degenerative disorders [28]. Embryonic stem cell (ESC)- and induced pluripotent stem cell (iPSC)-derived RPE cells are mainly used for visual restoration [29]. The methods have procedural difficulties and risk factors such as graft rejection and tumor formation [30]. In particular, differentiated RPE transplantation has stringent controls associated with retinal detachment [31]. To overcome these limitations, mesenchymal stem cell (MSC)-based therapy has become a promising strategy to restore retinal functions through various secreted factors, resulting in antiapoptotic and anti-inflammatory effects [32]. In patients with retinitis pigmentosa, implantation of umbilical cord-derived MSCs (UC-MSCs) into the suprachoroidal area resulted in the best corrected visual acuity and visual field after 6 months of follow-up in a phase 3 trial [33]. However, the mechanism of MSC therapy is still unknown. Consequently, the next generation of MSCs expressing PEDF was generated to examine the efficacy of MSC-based therapy based on our previous reports [23]. Compared with naïve PD-MSCs, PD-MSCs^PEDF^ highly secreted PEDF and augmented mitochondrial biogenesis and translation, as well as improved respiratory metabolism (Figure 1).

Since the main function of the retinal pigment epithelium is to structurally maintain photoreceptors, RPE degeneration results in the dysfunction of photoreceptors, leading to the loss of visual cycles. Photoreceptors depend on the retinal pigment epithelium for numerous metabolic functions, including the visual cycle. Typically, visual cycles regenerate 11-cis retinal through multiple steps involving specialized enzymes, such as retinoid binding protein (IRBP), and begin in the outer segment with all-trans retinal release from the opsin [12]. Therefore, the photoisomerization of 11-cis retinal to all-trans retinal in rod and cone photoreceptors is the initial step in vision induced by light, but the continued function of photoreceptors demands that all-trans retinal be converted into 11-cis retinal through the visual cycle.

In damaged RPE and photoreceptor cells, the autophagic mechanism converges for the recovery of vision loss. Zhou et al. reported that ATG5 knockdown in photoreceptors resulted in retinal degeneration and the deterioration of phototransduction protein levels [34]. During light-exposed photoreceptor degeneration, the deletion of Beclin1 or ATG7 in rod photoreceptor cells induced retinal degeneration via Parkin. Moreover, ABCA4- and RDH8-knockdown mice exhibited increased LC3B conversion, leading to the loss of photoreceptor cells [35]. Based on previous evidence, RDH11 expression in H_2_O_2_-injured rat retinas showed similar patterns to those of LC3B and PINK1 in our study (Figure 4). In RPE cells stimulated with glucose and diabetic retinas, decreased PEDF levels reduced the expression of RPE65, LRAT, and RDH11, which are visual cycle enzymes [36], and this effect occurred in AMD patients [37]. Similarly, our results indicate that H_2_O_2_-injured RPE cells cocultured with PD-MSCs^PEDF^ exhibited markedly increased RDH5, RDH11, LRAT, and RBP1 expression compared with those of cells cocultured with naïve PD-MSCs. Interestingly, the synthesis and secretion of PEDF by RPE cells were apically distributed to the retinal pigment epithelium around the outer segments of photoreceptors [38].

The pathogenesis of retinal degenerative diseases, including AMD and diabetic retinopathy, is highly associated with mitochondrial dysfunction, oxidative stress, and mitochondrial DNA damage [2]. Our previous study reported that PD-MSCs^PEDF^ promoted antioxidant defenses against mitochondrial ROS levels and improved mitochondrial biogenesis in oxidative stress-injured RPE cells as well as mitochondrial translation (Appendix A). Moreover, in an acute retinal degenerative rat model injured by H_2_O_2_, PD-MSC^PEDF^ transplantation had a prominent effect on mitochondrial dynamics compared to that of naïve PD-MSC transplantation, resulting in the recovery of retinal functions [22]. The coordination of mitochondrial biogenesis and mitophagy regulates mitochondrial contents and metabolism, preserving homeostasis [39]. Mitochondria impairment in RPE degeneration and the pathology of AMD induce a cellular defense mechanism known as mitophagy [19].

In the present study, we focused on facilitating mitophagy (mitochondrial autophagy) through PD-MSC^PEDF^ transplantation in H_2_O_2_-injured retinas. In cardiomyocytes, PEDF expression improved mitophagy to protect against hypoxic conditions by modulating the PEDF receptor (PEDF-R)/protein kinase C alpha (PKC-α) signaling pathway [40]. Although increased PKC-α expression by PEDF activates AMPK, which contributes to mitophagy in cardiovascular diseases [41], the effect of PEDF on mitophagy in the context of RPE degeneration is still unknown. We determined that the administration of PD-MSCs^PEDF^ augmented the expression of both the autophagosome marker LC3 and the mitophagy marker PINK1 in H_2_O_2_-induced rat retinas (Figure 5). Of the 10 retinal layers, LC3B expression was shown in ganglion cells (GCs), the inner plexiform layer (IPL), the inner nuclear layer (INL), the outer plexiform layer (OPL), photoreceptor inner segments (ISs), and pigment epithelial cells (PECs) [42]. In H_2_O_2_-injured rat retinas, these expression levels were observed in more than six layers (Figure 5). Interestingly, RPE cells cocultured with PD-MSC^PEDF^ exhibited significantly enhanced mitophagy, as indicated by the colocalization of PINK1 and the mitochondrial marker mtTracker compared to that of cells cocultured with naïve PD-MSCs. In addition, naïve PD-MSC coculture facilitated autophagy, but it did not induce mitophagy (Figure 6). These data suggest that PEDF could be highly relevant to mitophagy in H_2_O_2_-injured rat retinas.

Oxidative stress-induced experimental animal models have been reported in various methods [43,44,45,46,47]. Using transgenic mice, exon 3 of the antioxidant enzyme superoxide dismutase 2 (SOD2) was flanked, and confirmed to induce mitochondrial oxidative stress levels and the death of photoreceptor cells. However, transgenic methods have a longer time of 9 months, and do not induce drusen-like deposits [48]. In addition, radiation exposure can cause retinal damage and impaired vision. This method has several disadvantages, including cataracts and permanent blindness [49]. Especially, intravitreal injection using H_2_O_2_ can induce acute retinal degeneration against oxidative stress [21]. Additionally, H_2_O_2_-induced animal models have many benefits, such as taking a short time and having consistency, resulting in elevated oxidative stress and cellular apoptosis.

Previously, we confirmed therapeutic effects by PD-MSCs^PEDF^ in RPE cells and an acute retinal degenerative model. However, H_2_O_2_-injured rats acquired cataracts, and albino rats were known to have weak vision and be less resistant to stress levels [50]. For further study, we need to further establish chronic degenerative models, including streptozotocin (STZ)-induced diabetic or transgenic models, and investigate their molecular mechanisms.

## 5. Conclusions

In conclusion, the present study investigated whether functionally enhanced PD-MSCs^PEDF^ stimulated mitochondrial biogenesis and mitochondrial translation, as well as mitophagy for cellular clearance, resulting in the recovery of the visual cycle. Therefore, functionally enhanced PD-MSCs may be a new strategy for next-generation stem cell therapy for retinal degenerative diseases.

## Figures and Tables

**Figure 1 cells-10-01117-f001:**
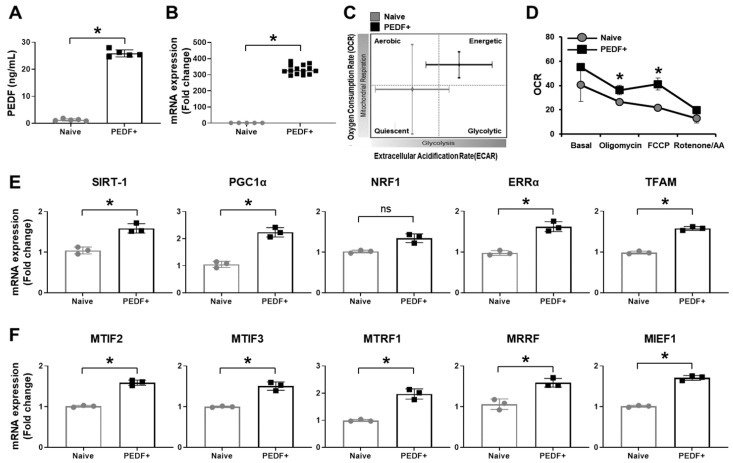
PD-MSCs^PEDF^ enhance mitochondrial activity. (**A**) Secreted PEDF in the cell culture supernatants of naïve PD-MSCs (Naïve) and PD-MSCs^PEDF^ (PEDF+) was analyzed by ELISA. OD = 450 nm. (**B**) The mRNA expression of PEDF was measured by qRT-PCR. (**C**,**D**) The OCR and ECAR levels of live Naïve and PEDF+ were analyzed by sequential treatment with 1 μM oligomycin, 0.5 μM FCCP, and 0.5 μM rotenone/AA by a Seahorse XF24 real-time analyzer. qRT-PCR analysis of (**E**) mitochondrial biogenesis markers (e.g., SIRT-1, PGC1α, NRF1, ERRα, and TFAM) and (**F**) mitochondrial translation markers (e.g., MTIF2, 3, MTRF1, MRRF, and MIEF1) in Naïve and PEDF+. The data from each group are shown as the mean ± SD and were analyzed by Student’s *t*-test. * *p* < 0.05 vs. Naïve.

**Figure 2 cells-10-01117-f002:**
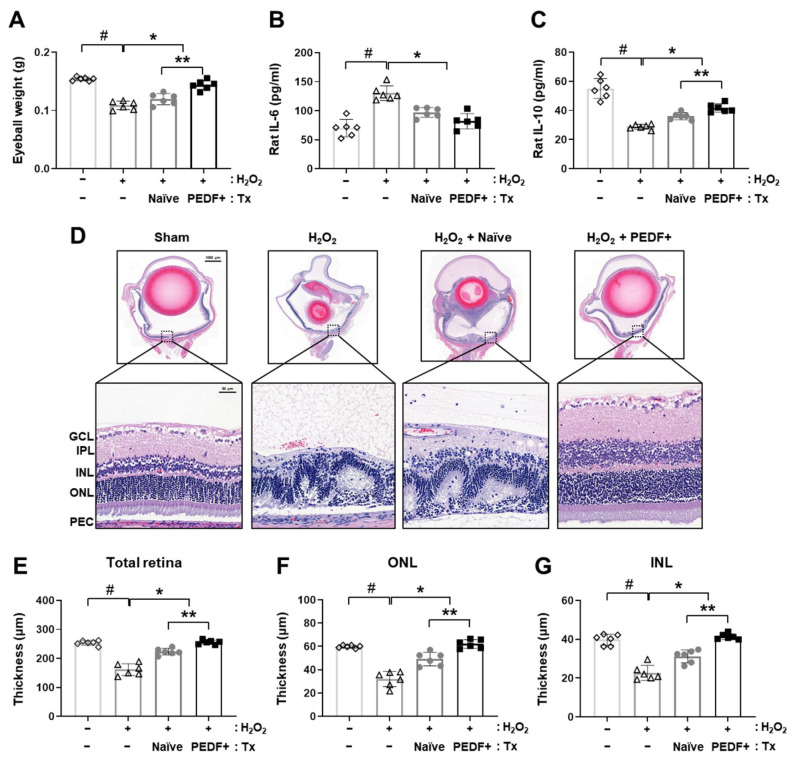
PD-MSC^PEDF^ transplantation restores retinal function in a H_2_O_2_-injured rat model. (**A**) Eyeball weight (gram; g) in each group (*n* = 6/group). We divided the rats into the following groups: sham, H_2_O_2_ injection (H_2_O_2_), intravitreal transplantation of naïve PD-MSCs (H_2_O_2_ + Naïve), and PD-MSCs^PEDF^ (H_2_O_2_ + PEDF+). (**B**) The levels of the proinflammatory cytokine IL-6 and (**C**) anti-inflammatory cytokine IL-10 in rat serum from whole blood (*n* = 6/group). (**D**) Histological images of H&E-stained rat eyeball sections. Measurement of (**E**) the total retina, (**F**) outer nuclear layer (ONL), and (**G**) inner nuclear layer (INL) in H&E-stained sections (*n* = 6/group). The data represent the mean ± SD and were analyzed by one-way ANOVA. # *p* < 0.05 vs. Sham (−), * *p* < 0.05 vs. H_2_O_2_, ** *p* < 0.05 vs. H_2_O_2_ + Naïve.

**Figure 3 cells-10-01117-f003:**
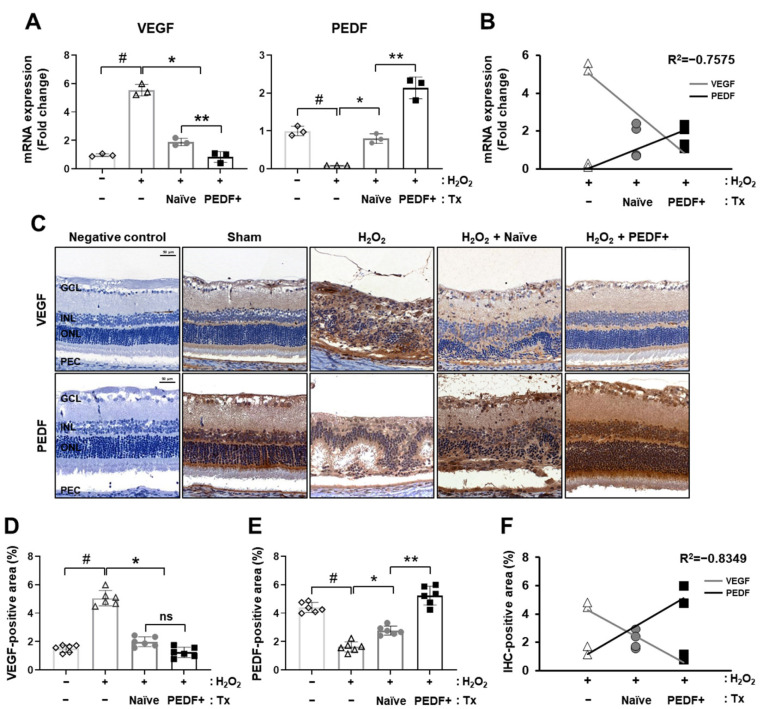
PD-MSCs^PEDF^ balanced VEGF and PEDF levels in a H_2_O_2_-induced rat model. The mRNA expression of (**A**) VEGF and PEDF in H_2_O_2_-injured rat retinas after naïve PD-MSC (H_2_O_2_ + Naïve) and PD-MSC^PEDF^ (H_2_O_2_ + PEDF+) transplantation. (**B**) The negative correlation between VEGF and PEDF was determined by qRT-PCR. (**C**) IHC staining and the intensities of (**D**) VEGF and (**E**) PEDF in the rat retina (*n* = 6/group). (**F**) The negative correlation between VEGF and PEDF was determined by the percentages of IHC-positive areas (%). The data represent the mean ± SD and were analyzed by one-way ANOVA. # *p* < 0.05 vs. Sham (−), * *p* < 0.05 vs. H_2_O_2_, ** *p* < 0.05 vs. H_2_O_2_ + Naïve.

**Figure 4 cells-10-01117-f004:**
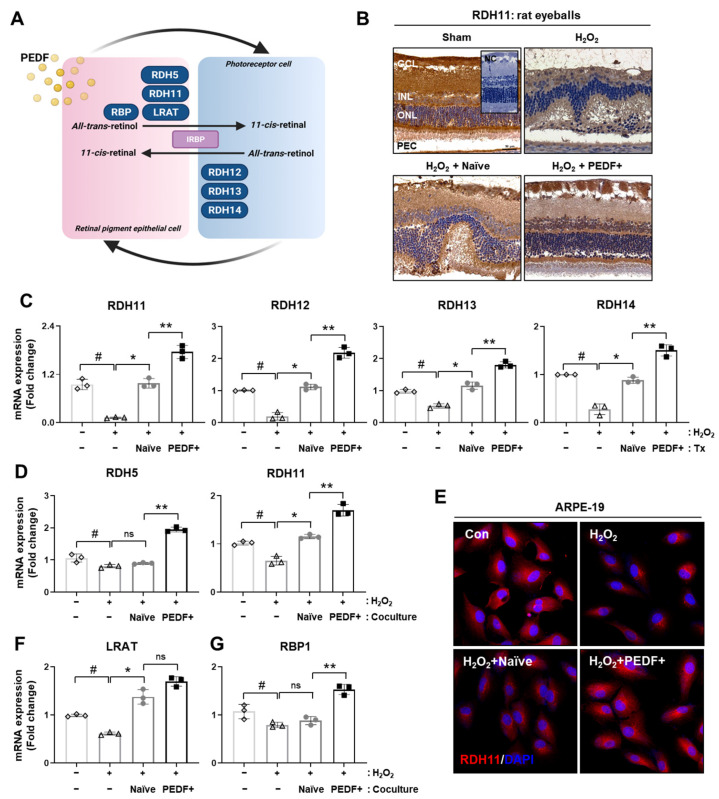
PD-MSCs^PEDF^ improved the visual cycles of rat retinal layers and RPE cells. (**A**) Summarized diagram of visual cycles in photoreceptor cells and RPE cells. (**B**) IHC staining showing RDH11 expression in H_2_O_2_-injured rat retinas after naïve PD-MSC (H_2_O_2_ + Naïve) and PD-MSC^PEDF^ (H_2_O_2_ + PEDF+) transplantation. (**C**) The mRNA expression of visual cycle genes in photoreceptors (e.g., RDH 12, 13, and 14) and RPE cells (e.g., RDH11) of each group (*n* = 6). # *p* < 0.05 vs. Sham (−), * *p* < 0.05 vs. H_2_O_2_, ** *p* < 0.05 vs. H_2_O_2_ + Naïve. (**D**) mRNA expression of the visual cycle of RPE genes (e.g., RDH5, RDH11, LRAT, and RBP1) in ARPE-19 cells. (**E**) Representative images showing RDH11 (red) in ARPE-19 cells by IF staining. DAPI (blue) was used for counterstaining. Scale bar = 100 μm. The mRNA expression of (**F**) LRAT and (**G**) RBP1 in ARPE-19. The data represent the mean ± SD and were analyzed by one-way ANOVA. # *p* < 0.05 vs. normal control (−), * *p* < 0.05 vs. H_2_O_2_, ** *p* < 0.05 vs. Naïve.

**Figure 5 cells-10-01117-f005:**
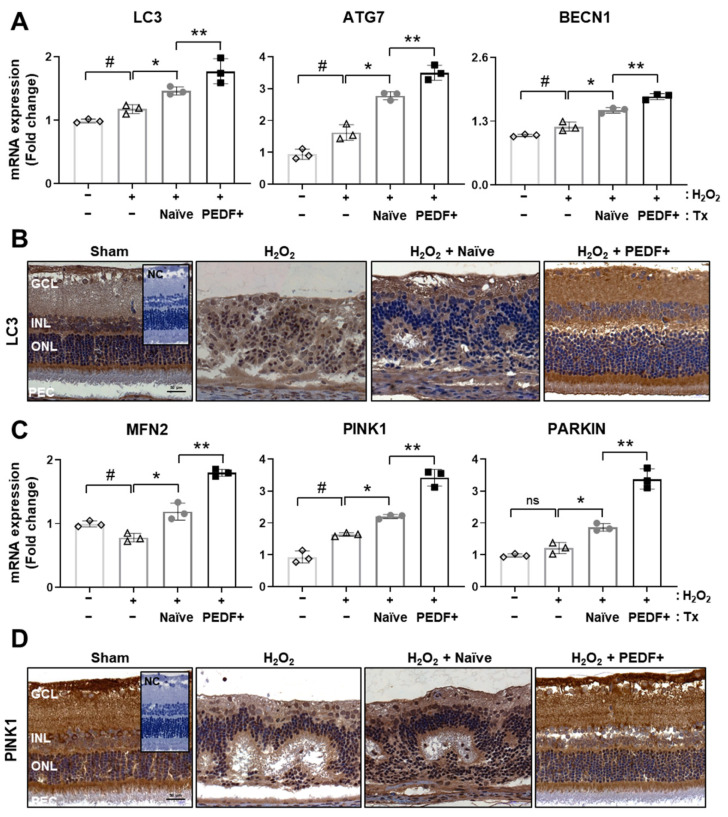
The administration of PD-MSCs^PEDF^ induces mitophagy in H_2_O_2_-injured rat retinas. (**A**) The mRNA expression of autophagy markers (e.g., LC3, ATG7, and BECN1) was determined by qRT-PCR after naïve PD-MSC (H_2_O_2_ + Naïve) and PD-MSC^PEDF^ (H_2_O_2_ + PEDF+) transplantation. (**B**) LC3 expression in the rat retinas of each group was determined by IHC (*n* = 6). Scale bar = 50 μm. (**C**) qRT-PCR analysis of mitophagy markers (e.g., MFN2, PINK1, and PARKIN) in the rat retina. (**D**) IHC analysis of PINK1 expression in the rat retina in each group (*n* = 6). Scale bar = 50 μm. The data represent the mean ± SD and were analyzed by one-way ANOVA. # *p* < 0.05 vs. Sham (−), * *p* < 0.05 vs. H_2_O_2_, ** *p* < 0.05 vs. Naïve.

**Figure 6 cells-10-01117-f006:**
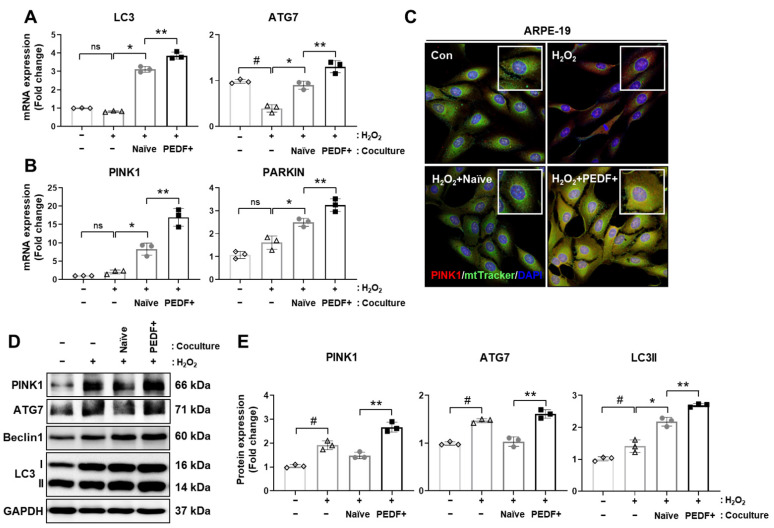
Coculturing RPE cells with PD-MSCs^PEDF^ enhances mitophagy. The mRNA expression of (**A**) autophagy markers (e.g., LC3 and ATG7) and (**B**) mitophagy markers (e.g., PINK1 and PARKIN) in H_2_O_2_-injured ARPE-19 cells cocultured with naïve PD-MSCs (H_2_O_2_ + Naïve) or PD-MSCs^PEDF^ (H_2_O_2_ + PEDF+) for 24 h was measured by qRT-PCR. (**C**) Immunostaining of PINK1 (red) and mtTracker (50 nM; green) in ARPE-19 cells. DAPI was used for counterstaining. Scale bar = 100 μm. (**D**) The protein expression and (**E**) intensities of autophagy and mitophagy markers in ARPE-19 cells measured by Western blotting. The data represent the mean ± SD and were analyzed by one-way ANOVA. # *p* < 0.05 vs. normal control (−), * *p* < 0.05 vs. H_2_O_2_, ** *p* < 0.05 vs. Naïve.

## Data Availability

Not applicable.

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
