# Peer review of "PEDF-Mediated Mitophagy Triggers the Visual Cycle by Enhancing Mitochondrial Functions in a H2O2-Injured Rat Model"

_cells, 2021, doi:10.3390/cells10051117_

Round 1
Reviewer 1 Report
This manuscript reported an interesting study that demonstrated PEDF-overexpressing MSCs prevent mitophagy, autophagy dysfunction, and oxidative damages to RPE and photoreceptors caused by H2O2 in the rat model and therefore restore visual cycle and function. Although interesting, some significant issues need to correct before considering for publication. First, all the IHC staining images are very poor, which makes them very difficult to appreciate and need to replace. Second, it is necessary to include visual function by ERG measurement to support the hypothesis. Third, since H2O2 induced a drastic increase of VEGF and decreased PEDF (Fig. 3), it is crucial to examine whether there are neovascularization and vascular leakage in this model. Lastly, the protein levels of autophagy marker LC3 II looked unchanged in the WB, but its quantification showed significant differences (Fig. 6).
Author Response
Dear, Dr. Editor,
We are pleased to submit the revision for article entitled “PEDF-Mediated Mitophagy Triggers the Visual Cycle by Enhancing Mitochondrial Functions in a H2O2-Injured Rat Model (Cells-1185730)”
We appreciate the positive and encouraging comments of the reviewers. Each of the answers to the questions has been sincerely written, and the revised parts of the thesis are marked in red.
We look forward to a positive response from reviewers.
With warm personal regards,
Gi Jin Kim, Ph.D.
Associate Professor
Dept. of Biomedical Science, CHA University;
689, Sampyeong-dong, Bundang-gu, Seongnam-si, Gyeonggi-do, Republic of Korea.
Tel: 82-31-881-3687, Fax: 82-31-881-4102, e-mail: [email protected]

Reviewer 2 Report
Manuscript cells-1185730 "PEDF-Mediated Mitophagy Triggers the Visual Cycle by Enhancing Mitochondrial Functions in a H2O2-Injured Rat Model" is a well-designed and well-conducted study on the effects of PEDF on retinal mitochondria and oxidative stress in the retina. I especially appreciate authors performing the Seahorse experiments to establish causality between PEDF and mitochondrial function. I don't have significant major criticism regarding the manuscript, but I would like to question the choice of H2O2 intravitreal injection. I understand the benefits of this model to be fast and consistent, and the author's data is conclusive; however, why authors choose the oxidative retinal damage model so detached from the clinical field? Transgenic oxidative stress models, such as SOD1 mice or NRF2 mice, would be way closer to the translational application of the findings and deserve the follow-up study. Authors need to address in their discussion the rationale for the H2O2 model choice (i.e., rat transgenic models of oxidative stress are limited). It appears that the author's group has established rat in vivo work rather than the mouse.) Add a statement of limitation addressing H2O2 has limitation as an oxidative damage model. Also, please add more background for the H2O2 injection model in the introduction.
Minor points:
- It would be easier to follow the antibody information is presented in the table form.
- All graphs should be updated to represent individual data points, not only standard deviation.
- All western blot results should be presented as whole uncropped membrane images in the supplementary data with the crop area presented in the main manuscript marked.
- Authors advised using black and white patterns in the graph to better differentiate experimental groups when manuscript printed in black and white as well as colorblind readership. The red and green pallet is a poor choice in this context.
- The use of H2O2 + Naïve is confusing as Naïve typically used for unaffected negative control animals; consider using the different designation, i.e., PD-MSCs and PD-MSCs PEDG.
- Add rationale for using DAB staining, i.e., strong photoreceptors autofluorescence.
- Fig 6. D/E, please clarify the fold change for LC3 I/II was calculated as the ratio of LC3 II to GAPDH or LC3 II to LC3 I?
- Interestingly administration of PD-MSCsPEDF rearrange retinal layers - please revise it appears intended meaning is "preserve the integrity of the retinal layers". The word "rearrange" in this context would mean that one retinal layer would change its place with another.
- Add experimental endpoint euthanasia in more detail.
- In 2.8 ELISA please provide the information on the plate reader and if reference wavelength was used to subtract the background.
Technical comment.
Not a criticism, but it appears that authors ARPE-19 cells of a low differentiation state. For future work, consider fully differentiating AEPE-19 by switching from DMEM:F12 to N1 media after confluence and culturing in N1 for at least 2 weeks. Usually, you will see more dramatic effects with H2O2 treatment using fully differentiated hexagonal ARPE-19, especially when cell edges are counterstained with ZO-1.
Author Response

(The authors gave the same response as above.)

Reviewer 3 Report
Brief Summary
The article by Kim and colleagues entitled ‘PEDF-Mediated Mitophagy Triggers the Visual Cycle by Enhancing Mitochondrial Functions in a H2O2-Injured Rat Model’ investigated whether functionally enhanced placenta-derived mesenchymal stem cells (PD-MSCs) with pigment epithelium-derived factor (PEDF) have stimulated mitochondrial biogenesis and mitochondrial translation that could help in retinal diseases. The authors also state that functionally enhanced PD-MSCs may be a new strategy for next-generation stem cell therapy for retinal degenerative diseases.
In general, the aims of the study are important. Using enhanced PD-MSCs could be a successful approach to heal retinal degeneration of any kind. With the rise in different retinal disease incidences like AMD we will not only need a better understanding of the disease but new approaches to help regeneration. However, the aims are clear, the presentation of the study is not and many details also need to be clarified. The authors used a diverse methodology to provide the necessary background for the results presented in this article but the description of them is mostly superficial. I am not convinced that the experiments presented here could be reproduced properly by others as they are missing many details, not to mention, animal experiments are based on albino rats. Also, the results are hard to interpret based on the insufficient approach. The details of the treatment, transplantation of the animals, and the handling of the cells are gappy and need to be described in detail. On top of this, the measurements are also inadequately described and presented.
If the authors could improve the transparency and reproducibility of the article that would be most welcomed and could make the article much more focused. These clarifications and the improvement of the conclusions should be done before proceeding in order to show how enhanced PD-MSCs could improve retinal health, where true regeneration rarely happens.
For details please see the specific comments here:
Specific comments
- I do not think that the following sentence strictly belongs to the abstract: ‘Previously, we reported that the overexpression of pigment epithelium-derived factor (PEDF) in placenta-derived mesenchymal stem cells (PD-MSCs; PD-MSCsPEDF) improved mitochondrial biogenesis and antioxidant effects in RPE cells in an acute retinal degenerative model caused by hydrogen peroxide (H2O2).’ I was unable to put it in context for the first reading, therefore, it does not add up to improve the understanding of the article.
- The abstract is full of abbreviations. E.g. ‘Compared to naïve PD-MSCs, PD-MSCsPEDF augmented mitochondrial biogenesis (e.g., PGC1α, 19 NRF1, ERRα, and TFAM) and translation (e.g., MTIFs, MTRF1, MRRF, and MIEF1) as well as mitochondrial respiratory states. -Please try to clarify the main findings here and put specific results in the results section and try to avoid using abbreviations.
- A list of abbreviations could be very useful for ease of understanding.
- ‘Loss of the visual cycle and photoreceptor damage in AMD are the major causes of blindness due to mitochondrial dysfunction.’-Please clarify. I am not sure being mitochondrial dysfunction the major cause of either wet or dry AMD, although it might be part of the problem.
- Sprague-Dawley rats are albino. It is well known that albinos have worse vision and could have altered retinas and less resistant to stress (eg. 10.1167/iovs.11-7602, 10.1371/journal.pone.0158082).
- ‘Acute eye disease was induced by intravitreal injection of H2O2 (10 μg/μl; Sigma-Aldrich) for 2 weeks…(L102.)’ -How many µl-s and what was the exact frequency/day? Were the animals anesthetized? Please, be more specific to aid reproducibility.
- Were the rat eyeballs open before submersion fixation? The quality of fixation depends on the accessibility of the retina for fixative.
- It is not clear how were the sections assorted for each antibody during the IHC. Please clarify it.
- From which region the sections belonged? Central, mid-central, peripheral? Although rats have less centralized retinas, they do have a level of eccentric cell-density difference.
- A table for the antibodies would be really useful, showing the types, cc. and ref. numbers of all primary and secondary antibodies used in IHC, WB, ELISA.
- Was GAPDH the endogenous control for each of the targets? Why did not the authors use the same primer set for GAPDH, PEDF, etc. (as indicated in Tables S1, 2)?
- In 2.6 WB.antibodies are not properly labeled (numbers, HRP-conjugated?) and the use of ImageJ is not appropriately described (plugin, endogenous ctrl., reference samples).
- 2.8 ELISA is missing enough detail to reproduce.
- Please label the exact statistical test used for each of the figures produced.
- In the results section, the authors begin with previously described results that might belong to the discussion to interpret the new results.
- What part of the data has been already published before (e.g. fig 5. In 10.1038/s41374-020-0470-z )?
- The figure parts are labeled with capital letters, while the figure captions are not.
- In Fig 2. the borders of ONL & INL are not clear in the H2O2 and H2O2 + Naive retinas. How it is even possible to measure them?
- In Fig 3.C the BG intensities are clearly not normalized.
- In Fig 3. the Sham control also shows high VEGF mRNA and IHC expression, while low PEDF mRNA and high IHC expressions. How can the authors explain this?
- How could co-culturing RPE cells with PD-MSCs(PEDF) relate to the effects on the whole, live retina?
- On Fig 6. C high magnification images could be useful to show cellular details.
Other comments
L49. that protect
L94. co-culture
L186. with the ‘OR’ a mitochondrial
L189. proliferator-activated
L190. estrogen-related
L260. showed the same trend
L293. of the visual cycle
L404. cells were measured
L458. cells were apically
L492. models
L492. the streptozotocin (STZ)-induced diabetic model -(as it induces DR as a secondary effect)
L497. the recovery
Author Response

(The authors gave the same response as above.)

Round 2
Reviewer 3 Report
The authors managed to reply to all points concerning this article, however, there are still some points I do not think were accurately updated to reach a higher level of understanding by the readers. I would like to highlight these points. After their correction, I see no more obstacles ahead to publish this article as it contains an interesting piece of data.
- I still do not think that this is correct: L102: 'Acute eye disease was induced by intravitreally single injection of H2O2 (10 μg/μl; Sigma-Aldrich) for 2 weeks, and rats in the sham group were injected with balanced salt solution (BSS).' -Maybe... 'Acute eye disease was induced by a single intravitreal injection of H2O2 (10 μg/μl; Sigma-Aldrich), while the sham group was injected with balanced salt solution (BSS). The animals were held for 2 weeks before sacrificed.' ? This would be crucial to see if they were injected daily for two weeks or once with a two-week survival period.
- This newly introduced sentence also has less sense: (L473)' However, H2O2-injured rats are concerned about an acquired cataract and albino rats were known as worse vision and less resistant to stress-levels [50]. Our studies are needed to further establish chronic degenerative model including streptozotocin (STZ)-induced diabetic models or transgenic models and investigate their molecular mechanisms.' -Please, check it with the help of a native speaker.
- For point #6. Please, add one sentence in the text to clarify the fixation method.
- What do you mean by your reply in point #8: 'We were used in the middle of each rat eyeball.'? -Were these parts from the central (arround the optic-disc), mid-central (halway between the OD and the ora serrata) or peripheral regions in the retina? It is important to match the samples, otherwise the authors may get false conclusions.
- Point #17. Could you give a better explanation or an example as a Supplementary figure by labeling the layers and measurement sites?
- Point #18. How many sections and what is the exact sofware ( histoquantification program in a 3DHISTECH)? Could you add this part in M&Ms?
Author Response
Cover letter
Article Submission in Cells
Dear Reviewer,
We greatly appreciate your careful evaluation of our 1st revised manuscript (Cells-1185730) entitled: “PEDF-Mediated Mitophagy Triggers the Visual Cycle by En-hancing Mitochondrial Functions in a H2O2-Injured Rat Model.” We were really encouraged by the reviewers’ positive comments and constructive suggestions. I am happy to report that we have successfully addressed all issues and concerns through additional data and subsequent revision of our manuscript, as detailed in the following response page. As Reviewer’s commented, we corrected it clearly stating with each comment and changes are highlighted in red in the revised manuscript,
In summary, based on the insightful and constructive criticisms provided by both referees, we believe that our manuscript is significantly improved and we hope that you will consider it suitable for publication in Cells.
Very sincerely yours,
Gi Jin Kim, Ph.D.
Associate Professor
Dept. of Biomedical Science, CHA University;
689, Sampyeong-dong, Bundang-gu, Seongnam-si, Gyeonggi-do, Republic of Korea.
Tel: 82-31-881-3687, Fax: 82-31-881-4102, e-mail: [email protected]
